# Utilization of companionship during delivery and associated factors among women who gave birth at Arba Minch town public health facilities, southern Ethiopia

**Kassaw Beyene Getahun, Gebresilasea Gendisha Ukke, Biresaw Wassihun Alemu** *

Department of Midwifery, College of Medicine and Health Science, Arba Minch University, Arba Minch, Ethiopia

☯ These authors contributed equally to this work.
* bireswas@gmail.com

## Abstract

### Background

Companionship during delivery is an important feature of compassionate and respectful maternity care. It has a positive impact on delivery and birth outcomes. In low resource countries like Ethiopia lack of companionship discourages women from accessing facility-based delivery care. Therefore, this study aimed to assess the utilization of companionship during delivery and associated factors.

### Methods

Health facility-based cross-sectional study design was done from October to November 2019. Interviewer administered questionnaires were used to collect the data from 418 study participants. The data were entered with Epi data version 4.4 and exported to Statistical Package for Social Sciences (SPSS) version 25.0 for analysis. Binary logistic regression was done. Statistical significance was declared at P- values < 0.05 with a 95% confidence level.

### Results

The finding of the study showed that only 13.8% of mothers utilize companionship during delivery. Variables such as having a desire to have companionship during delivery in the health facilities (AOR = 5.17, CI 95% 2.63, 10.16), having complication during the labor and delivery (AOR = 3.48, CI 95%, 1.81, 6.70), and being primipara (AOR = 2.05, CI 95% 1.09, 3.87) were the independent factors associated with companionship utilization.

### Conclusions

The finding of the study showed that the utilization of companionship during delivery was low. Permitting women to have a companion of choice during labor and childbirth can be a cost-effective intervention to improve the quality of maternity care, facing complications

**Data Availability Statement:** All relevant data are within the manuscript and its Supporting Information files.

**Funding:** The authors received no specific funding for this work.

**Competing interests:** The authors declare that they have no competing interests.

**Abbreviations:** AGH, Arbaminch General Hospital; ANC, Antenatal Care; CDD, Companionship during Delivery; RMC, Respectful Maternity Care; SSA, Sub Saharan Africa; WHO, World Health Organization.

during delivery, having a desire to have companionship during delivery and primiparous women were more likely to utilize companionship. To improve this low utilization of companionship institutions and care providers should provide information about companionship during antenatal care attendance. Besides, there is a need for clear guidelines to govern the practice of companions.

## Background

Worldwide, around 140 million births occur every year and the majority of these are vaginal births and most of these occur without complications for women and their babies [1]. However, in situations where complications arise during labor, the risk of serious morbidity and death increases for both the woman and baby. Over a third of maternal deaths and a substantial proportion of pregnancy-related life-threatening conditions are attributed to complications that arise during labor, childbirth, or the immediate postpartum period [2, 3]. And the majority of stillbirths and neonatal deaths result from complications during labor and childbirth [4]. The burden of maternal and perinatal deaths is suspiciously higher in low resource countries compared to developed countries, Therefore, improving the quality of care around the time of birth, especially in low-income countries, has been identified as the most vital strategy for reducing maternal and newborn death [5, 6].

Labour companionship refers to support provided to a woman during labor and childbirth. It may be provided by a partner, family member, friend, doula, or healthcare professional [7]. There is a global interest in improving the quality of maternal and newborn care [8]. Labor companionship is a key component of providing respectful maternity care and has been recommended most recently as part of WHO recommendations on intrapartum care for positive childbirth experience and included as one of the standards for improving the quality of maternal and newborn care in health facilities [9–11].

World health organization (WHO) recommends that facilitating and ensuring clear and respectful communication between health-care providers and the woman in labor, and providing continuous emotional support is advocated for all women. However, in the actual clinical setting, it not well-practiced [12, 13]. Permitting and supporting the presence of a woman's companion of choice during labor and childbirth is vital to reduce mistreatment or abuse in a health facility [13]. Studies have shown that women who receive continuous support during labor are more likely to deliver spontaneously and Therefore require fewer cesarean sections or operational deliveries; have a shorter length of labor; are less likely to require intrapartum analgesia; are more satisfied with their childbirth experience, and are less likely to have a baby with a low five-minute Apgar score [14]. In addition to benefiting women in labor, companions may also help reduce staff workload and improve processes. Companions may allow health staff to attend to urgent issues, remind staff when it is time to re-examine women or when there is a sudden change, arrange transportation if complications arise, and reinforce messages and instructions to women [15].

One of the rights specified in the Respectful Maternity Care Charter (RMC) is "respect for her choices and preferences, including companionship during maternity care. but it is a neglected area [16, 17].

A previous study showed that the low practice of labor companions was associated with the absence of guidelines, lack of infrastructure to protect privacy, overcrowding of ward and poor knowledge and negative attitudes of health-care providers [18, 19].

Now a day, companionship being increasingly recognized as an integral component of respectful maternity care and a potentially important factor in facility delivery rates, a paucity of evidence exists on the factors that predict it [19, 20].

In Ethiopia, companionship during delivery is not well studied. Therefore, this study is aimed to assess the utilization of companionship during childbirth and associated factors among women who give birth at Arba Minch town public health facilities, South Ethiopia.

## Methods

### Study setting and design

A health facility-based cross-sectional study design was carried out in Arbaminch town public health facilities from October to November 2019. Arbaminch town is the administrative city of the Gamo zone, southern Ethiopia, which is 454km south of Addis Ababa (the capital city of Ethiopia) and about 280 Km from Hawassa (the capital of SNNP). The town is subdivided into 4-sub city and 11 kebeles (the smallest administrative structure in Ethiopia). The town has a total area of 5556 hectares and a total population of 112,724 among those (50.2%) of them were females. The number of health institutions in Arba Minch town is 1 governmental general hospital, 2 health centers, 33 private clinics, 12 drug store, and 2 community pharmacy.

### Populations

**Source population.**   All women who gave birth at Arba Minch town public health facilities.

**Study population.**   All women who gave birth in Arba Minch town public health facilities during the study period.

### Inclusion and exclusion criteria

**Inclusion criteria.**   All women who were laboring and gave birth at Arba Minch town public health facilities.

**Exclusion criteria.**   Those women who are seriously ill and unable to communicate during the data collection period were excluded from the study.

### Sample size determination

The sample size was calculated using a single population proportion formula by considering the following assumptions: 95% confidence level, the margin of error (0.05), p = 44.7% [21]. The required sample size after adding a 10% non-response rate was 418.

### Sampling techniques & procedure

There is one public hospital (Arbaminch general hospital) and two public health centers (Sikela and Shecha health centers) in Arbaminch town and all were included in the study. The allocation of the sample to health facilities was made proportionally based on the number of women who give birth at each facility in the two months preceding the data collection period.

Individual study subject at each health facility was selected by systematic random sampling during the data collection period until the required sample size at each health facility was obtained.

The sampling interval k = 2 was calculated by dividing the source population to the total sample size and this interval was used in all health facilities to select study subjects.

Therefore, the first women from each health institution were selected by lottery method. Then every other woman from each health institution was interviewed.

## Operational definitions

**Labor companionship.** Support provided to laboring women at all moments of the labor process. It may be provided by a partner, family member, or social network [22].

**Utilization of companionship.** Having a support person of laboring women to provide support and stay with her during labor in the health facilities.

## Data collection tool and quality control

Before actual data collection occurred two-day training was provided for data collectors and the supervisor about techniques of data collection and briefed on each question included in the data collection tool. The pretest was done on 5% [21] of mothers receiving care in a health center that was not included in the study before the actual study period. After pre-testing the questionnaire, Cronbach's Alpha was calculated by using SPSS window version 25.0 to test internal consistency (reliability) of the item, and Cronbach's Alpha greater than 0.7 was considered as reliable. Data were collected by trained midwives and nurses. During data collection, regular supervision was done by the supervisors.

## Data analysis and interpretation

The collected data were checked manually for completion and any incomplete or misfiled questions, cleaned and stored for consistency, entered into EpiData version 4.4. (EpiData Association, Odense, Denmark), and then exported to SPSS version 25.0 (IBM Corp., Armonk, NY, USA) for analysis. Descriptive statistics were calculated and presented using tables and figures. Multivariable logistic regression analysis was performed to adjust for possible confounding variables. Variables that were significant in the bivariate logistic regression were entered into the multiple regression analysis. The $p < 0.05$ or 95% confidence intervals (CIs) not including 1.0 were considered to indicate statistical significance.

## Ethical approval and consent to participant

Ethical clearance was obtained from the institutional Research Ethics review board of the college of medicine and health science, Arba Minch University. Permission was obtained from the managers of each health facility. After the purpose and objective of the study have been informed, informed verbal consent was obtained from each study participant. Moreover, the confidentiality of information was guaranteed by using code numbers rather than personal identifiers and by keeping the data locked. Data were collected before discharge to home after she was stable and comfortable.

## Results

### Socio-demographic characteristics of respondents

Four-hundred seven women participated in the study with a 97.3% response rate. The mean age of the study participants was 26 years (SD± 4.86 years) and 181 (44.5%) women were between the age group of 25–34 years. 281 (69%) of respondents were urban residents and 386 (94.8%) of the study participants were married. Among the total respondents, 146 (35.9%) of women had a primary education level and One hundred ninety-seven (48.4%) were Orthodox Christians. Half of the respondents were from Gamo ethnic group (51.4%) followed by Gofa ethnic group 68 (16.7%) (Table 1).

**Table 1. Socio-demographic characteristics of the study participants, Arba Minch town, south Ethiopia, 2019, (n = 407).**

| Variables | Frequency | Percentage |
|---|---|---|
| Age | | |
| <25 | 171 | 42 |
| 25–34 | 181 | 44.5 |
| ≥35 | 55 | 13.5 |
| Residency | | |
| Rural | 126 | 31 |
| Urban | 281 | 69 |
| Marital status | | |
| Married | 386 | 94.8 |
| Single | 11 | 2.7 |
| Divorced | 8 | 2 |
| Widowed | 2 | 0.5 |
| Occupation | | |
| Housewife | 212 | 52.1 |
| Government employee | 69 | 17.0 |
| NGO/private | 99 | 24.3 |
| Others | 27 | 6.6 |

Key = *traditional, Jehovah witness, **Konso, Derashe, Gurage, Amaro, ^ Students, Daily labor.

## Obstetrics characteristics of the respondents

Two hundred thirty (56.5%) of the study participants were multiparous and 54 (29.3%) had labor companionship during previous institution delivery. Almost all 363 (89.2%) women had antenatal follow up during current pregnancy and only 45 (12.4%) of women had got information from health care providers about labor companionship during antenatal care attendance.

The majority of 346(85%) of respondents perceived that allowing laboring women to have a companion during childbirth. Of the total respondent, 387(95.1%) of them had planned pregnancy (Table 2).

## Benefits of companionship during delivery

Majority of respondent 77.2% mention that having companionship during delivery can reduce loneliness followed by reducing labor pain management (Fig 1).

## Utilization of companionship during delivery

Of the total respondents, 56 (13.8%) of laboring mothers utilize companionship during delivery, and 351(86.2%) do not utilize companionship. The main reason mentioned for not to utilize companionship during delivery was provider denial 47.9% followed by an institution not allowed 21.1% (Fig 2).

## Factors associated with having a companion during delivery

To determine the association between utilization of companionship during delivery in the health facilities with different factors, the following dependent variables were checked against outcome variables. On bivariate analysis, women's occupation, family monthly income, complication during labor and delivery, parity of woman, Comfortability of facilities to be

**Table 2. Obstetrics characteristics of women who give birth in Arbaminch town public health facilities southern Ethiopia, 2019.**

| Variables | Frequency | Percentage |
|---|---|---|
| Parity | | |
| Primipara | 177 | 43.5 |
| Multipara | 230 | 56.5 |
| Have companion during the last delivery (n = 230) | | |
| Yes | 54 | 29.3 |
| No | 130 | 70.7 |
| Place of last delivery (n = 230) | | |
| Home | 48 | 20.9 |
| Health institutions | 182 | 79.1 |
| Did you attend ANC in your current pregnancy (n = 407) | | |
| Yes | 363 | 89.2 |
| No | 44 | 10.8 |
| Did the provider inform you about companion (n = 363) | | |
| Yes | 45 | 12.4 |
| No | 318 | 87.6 |
| Status of Pregnancy | | |
| Planned | 387 | 95.1 |
| Unplanned | 20 | 4.9 |

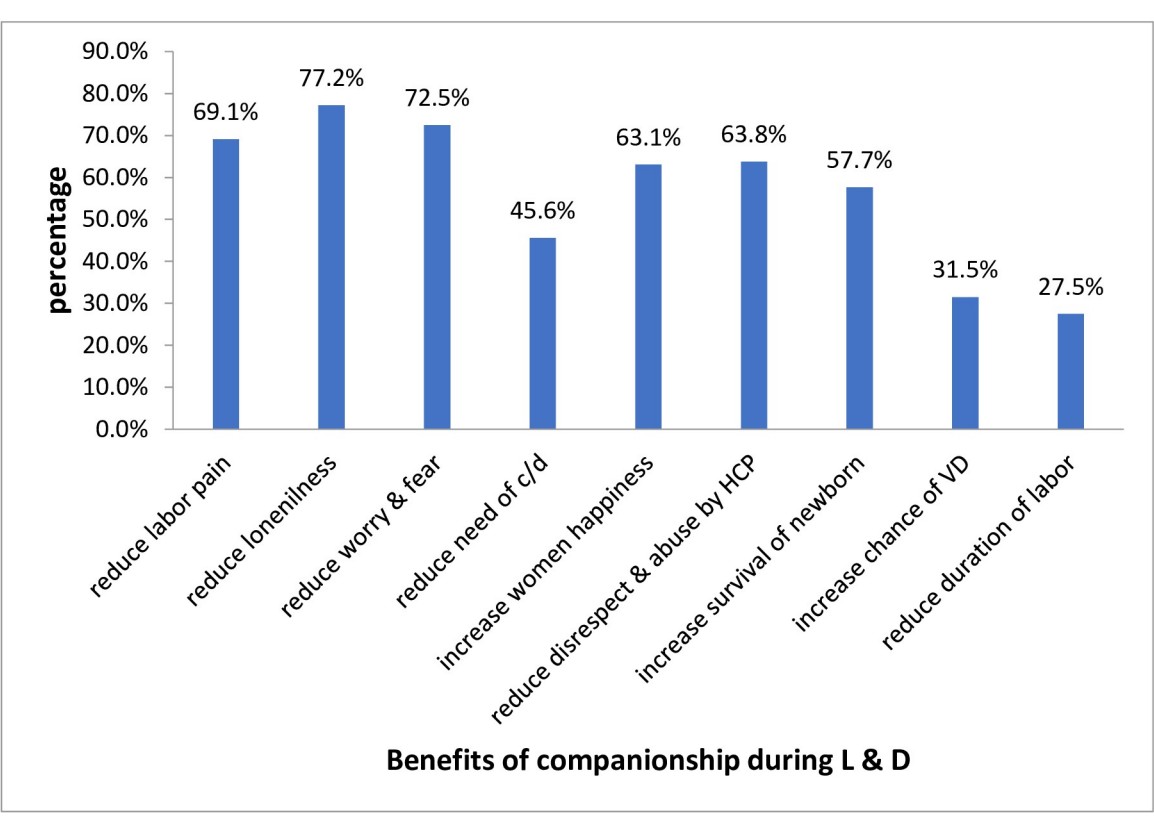

**Fig 1. Benefits of companionship during labor and childbirth mentioned by respondents in Arbaminch town public health facilities, south Ethiopia, 2019.**

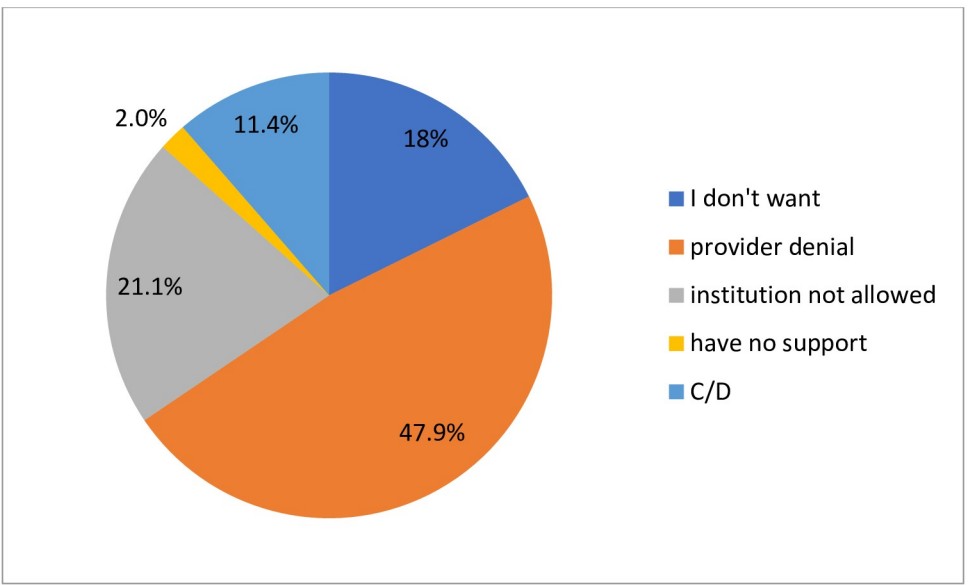

**Fig 2. The distributions of a reason not to utilize companionship during delivery in Arbaminch town public health facilities, south Ethiopia 2019.**

accompanied and knowledge had significantly associated with the utilization of companion during delivery in the health facilities.

After controlling the effects of confounder on multivariable analysis, having a desire to be accompanied, and complication during labor and delivery and parity have a statistically significant association with utilization of companionship during delivery. Respondents who had a desire to be accompanied during labor and delivery were 5 times more likely to be accompanied by their companion than others (AOR = 5.17 (2.63, 10.16). Those respondents who have had complications in the current pregnancy and labor were 3,48 times more likely to utilize their companionship than others (AOR = 3.48 (181, 6.70). Besides, those respondents who gave birth for the first time (primipara) were 2.05 times more likely to have been accompanied by their companion than multiparous women (AOR = 2.05, 1.09, 3.87) (Table 3).

## Discussions

In this study, the overall utilization of companionship during delivery was found to be 13.8%. The finding of this study is lower than the study done in Brazil 38.1%, and Kenya 67%, [23, 24]. The discrepancy might be due to the cultural difference in labor companion and policy that enforce health care providers to allow labor companion. Similarly, the finding of this study is lower than the other study conducted in Ethiopia, Kenya, Madagascar, Rwanda, and the United Republic of Tanzania which showed that the proportion of facilities that encouraged women to have a companion were 34%,38%,40%,43% and 67% respectively [25]. The finding of this study was similar to the study conducted in the Kigoma Region, Tanzania, which showed that only 12% of mothers companionship during delivery [15]. In contrast, this rate is higher than was reported in a study conducted in rural central Ghana which showed that 58% of mothers utilize companionship during delivery [26]. The finding of this study is lower than the study reported in Nigeria which showed that only 22.1% of mothers companionship during delivery [27].

**Table 3. Bivariate and multivariable analysis of factors associated with utilization of companion during delivery, Arbaminch Ethiopia, Feb 2019 (n = 407).**

| Variables | Having companion during delivery | | Odds Ratio with 95% CI | |
|---|---|---|---|---|
| | Yes | No | COR | AOR |
| Women occupation | | | | |
| Unemployed | 25 | 187 | 1 | 1 |
| Employed | 31 | 164 | 0.70(0.40,, 1.24) | 1.48(0.77, 2.78) |
| Monthly family income | | | | |
| ≥3000 ETB | 29 | 211 | 0.71(0.40, 1.25) | 0.52(0.27, 1.02) |
| <3000 ETB | 27 | 140 | 1 | |
| Desire to have a companion during delivery | | | | |
| Yes | 42 | 135 | 4.80(2.52, 9.12) | 5.17(2.63, 10.16)* |
| No | 14 | 216 | 1 | 1 |
| Complications during pregnancy & labor | | | | |
| Yes | 24 | 63 | 3.42(1.89, 6.21) | 3.48(181,6.70)** |
| No | 32 | 288 | 1 | 1 |
| Parity | | | | |
| Primipara | 35 | 142 | 2.45(1.37, 4.38) | 2.05(1.09, 3.87)* |
| Multiparous | 21 | 209 | 1 | 1 |
| the facility was comfortable to be accompanied | | | | |
| Yes | 15 | 70 | 1.46(0.76, 2.80) | 1.50(0.73, 3.08) |
| No | 41 | 281 | 1 | 1 |
| Knowledge | | | | |
| Good | 6 | 46 | 0.79(0.32, 1.96) | 0.84(0.31, 2.26) |
| Poor | 50 | 305 | 1 | 1 |

* = p-value <0.05,

** = p-value≤ 0.01, CI = Confidence Interval, COR = Crude Odds Ratio, AOR = Adjusted Odds Ratio.

The finding of this study showed that being primiparous (delivered for the first time) were two times more likely to be accompanied by their labor companion during childbirth in the health facilities than those women who were multiparous. This finding is similar to the study done in Brazil, rural central Ghana, and Kigoma Region, Tanzania, which revealed that being primiparous (delivered for the first time) were more likely to be accompanied by their labor companion during childbirth in the health facilities than those women who were multiparous [23, 15, 26]. This might be due to the fear of childbirth because most of the time primiparous women may face the fear of childbirth and they will be more likely accompanied by their companion and this fear of childbirth can harm a woman's psychological wellbeing and associated with adverse obstetric outcomes and postpartum mental health difficulties.

In this finding women who had obstetrics or medical complications during labor and delivery were 3.48 times more likely to be utilized labor companion as compared to those women who had never been experiencing any complications during labor and delivery. This result is supported by a study conducted in Tanzania which showed that laboring women who develop complications during childbirth had significantly greater odds of having companionship during delivery than women who had normal labor and delivery [21]. But in contrast, a study in Kenya [24] showed that women who had experienced complications at labor are 66% less likely to have companionship while giving birth in the health facilities. This difference may be encountered due to women with labor and delivery complication needs strict follow up by health care provider alone, to provide appropriate management without intervention, and to avoid additional stress by her family members.

## Conclusion

Supportive care is beneficial for women during labor as it positively affected pain perception and feelings of anxiety of the parturient. The finding of this study showed that the utilization of companionship during delivery was low as compared to the previous study. Some of the factors associated with the utilization of companionship during delivery was having a desire to companionship, being primiparous, and having facing complication during pregnancy and delivery. Allowing women to have a companion of choice during delivery can be a low-cost and effective intervention to improve the quality of maternity care. More women are now giving birth in health facilities, but the poor quality of care can put their lives and well-being—and that of their infants—at risk. It is therefore crucial to ensure that women and their newborn infants are provided with respectful, high-quality care throughout pregnancy and childbirth. One potential way to improve the quality of care during childbirth in health facilities may be for women to be continuously supported by another person throughout labor.

## Supporting information

**S1 File. Questionary on companionship during labor and childbirth.**
(DOCX)

**S1 Dataset. The SPSS data file on companionship during labor and childbirth.**
(SAV)

## Acknowledgments

The authors thank all the study participants and data collectors.

## Author Contributions

**Conceptualization:** Kassaw Beyene Getahun, Gebresilasea Gendisha Ukke, Biresaw Wassihun Alemu.

**Data curation:** Kassaw Beyene Getahun, Gebresilasea Gendisha Ukke, Biresaw Wassihun Alemu.

**Formal analysis:** Kassaw Beyene Getahun, Gebresilasea Gendisha Ukke, Biresaw Wassihun Alemu.

**Funding acquisition:** Kassaw Beyene Getahun, Gebresilasea Gendisha Ukke, Biresaw Wassihun Alemu.

**Investigation:** Kassaw Beyene Getahun, Gebresilasea Gendisha Ukke, Biresaw Wassihun Alemu.

**Methodology:** Kassaw Beyene Getahun, Gebresilasea Gendisha Ukke, Biresaw Wassihun Alemu.

**Project administration:** Kassaw Beyene Getahun, Gebresilasea Gendisha Ukke, Biresaw Wassihun Alemu.

**Resources:** Kassaw Beyene Getahun, Gebresilasea Gendisha Ukke, Biresaw Wassihun Alemu.

**Software:** Kassaw Beyene Getahun, Gebresilasea Gendisha Ukke, Biresaw Wassihun Alemu.

**Supervision:** Kassaw Beyene Getahun, Gebresilasea Gendisha Ukke, Biresaw Wassihun Alemu.

**Validation:** Kassaw Beyene Getahun, Gebresilasea Gendisha Ukke, Biresaw Wassihun Alemu.

**Visualization:** Kassaw Beyene Getahun, Gebresilasea Gendisha Ukke, Biresaw Wassihun Alemu.

**Writing – original draft:** Kassaw Beyene Getahun, Gebresilasea Gendisha Ukke, Biresaw Wassihun Alemu.

**Writing – review & editing:** Kassaw Beyene Getahun, Gebresilasea Gendisha Ukke, Biresaw Wassihun Alemu.

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
