## [Decision Letter · Decision Letter 0]

23 Jul 2020

PONE-D-20-18273

Utilization of Companionship during delivery and associated factors among women who gave birth at Arbaminch town public health facilities, southern Ethiopia

PLOS ONE

Dear Dr. Alemu,

Thank you for submitting your manuscript to PLOS ONE. After careful consideration, we feel that it has merit but does not fully meet PLOS ONE’s publication criteria as it currently stands. Therefore, we invite you to submit a revised version of the manuscript that addresses the points raised during the review process.

SPECIFIC ACADEMIC EDITOR COMMENTS: Two experts in the field reviewed your manuscript. We thank them for their time and efforts. Although some interest was found in your study, several major concerns overshadowed this enthusiasm. These concerns include: the article structure needs significant work with emphasis on stating the novelty of this study; questions about delivery route, how women perceived that they were allowed to have a birth companion, and the exclusion criteria; the data presentation needs to be stronger; and the discussion and conclusion need to better reflect the findings. All of the reviewers' comments must be addressed in your revised manuscript.

We look forward to receiving your revised manuscript.

Kind regards,

Frank T. Spradley

Academic Editor

PLOS ONE

2. Please include additional information regarding the survey or questionnaire used in the study and ensure that you have provided sufficient details that others could replicate the analyses. For instance, if you developed a questionnaire as part of this study and it is not under a copyright more restrictive than CC-BY, please include a copy, in both the original language and English, as Supporting Information.  If the original language is written in non-Latin characters, for example Amharic, Chinese, or Korean, please use a file format that ensures these characters are visible.

3. Please state whether you validated the questionnaire prior to testing on study participants. Please provide details regarding the validation group within the methods section.

5.Thank you for stating the following in the Funding Section of your manuscript:

[Arba Minch University as a requirement for postgraduate studies supports this research

financially.]

 [no]

Reviewers' comments:

Reviewer's Responses to Questions

**Comments to the Author**

1. Is the manuscript technically sound, and do the data support the conclusions?

Reviewer #1: Yes

Reviewer #2: No

2. Has the statistical analysis been performed appropriately and rigorously? 

Reviewer #1: Yes

Reviewer #2: I Don't Know

3. Have the authors made all data underlying the findings in their manuscript fully available?

Reviewer #1: Yes

Reviewer #2: No

4. Is the manuscript presented in an intelligible fashion and written in standard English?

Reviewer #1: Yes

Reviewer #2: No

5. Review Comments to the Author

Reviewer #1: While it is interesting to see the variables which are correlated with birth companionship, this piece could be made stronger with qualitative findings particularly since the paper reports that many women did not perceive that they were allowed to have a birth companion.

Reviewer #2: all comment inserted in manuscript and attached. some comments are including:

article structure is weak.structure paragraph is incorrect. major revision is needed.

background is low. what is the importance ,necessary,gap and research question?

What delivery type with normal vaginal delivery or cesarean delivery?

exclusion criteria is not completed.

please explain more about validity and reliability of questionnaire.

what is ethical code?

the table has ambiguity. some lines in table have to delete to clarify it.

discussion is weak and underdeveloped and do not cover all finding such as associated factors.

6. PLOS authors have the option to publish the peer review history of their article (what does this mean?). If published, this will include your full peer review and any attached files.

Reviewer #1: No

Reviewer #2: No

---

## [Author Response · Author response to Decision Letter 0]

5 Aug 2020

Author’s Point-by-Point Response to the Reviewer's and Editors Reports

Title: Utilization of Companionship during delivery and associated factors among women who gave birth at Arba Minch town public health facilities, southern Ethiopia: A cross-sectional study

Corresponding author: Biresaw Wassihun /bireswas@gmail.com

Authors

1. Kassaw Beyene

2. Gebresilasea Gendisha

3. Biresaw Wassihun

Manscurpuit number: PONE-D-20-18273

Journal: Plos one

Article type: Research article

Point by point response to Reviewers and Editors

First of all, the authors would like to thank Plos one Journal editors and the respective reviewers for reviewing our manuscript and providing the necessary comments to be corrected. As per the comments given, we have made corrections point by point to comment. The authors tried to answer all the issues raised by editorial team and reviewers. 

Point by point response to Editor

1. Please ensure that your manuscript meets PLOS ONE's style requirements, including those for file naming."" 

Response: Thank you very much we had apply journal requirement 

2. Please include additional information regarding the survey or questionnaire used in the study and ensure that you have provided sufficient details that others could replicate the analyses. For instance, if you developed a questionnaire as part of this study and it is not under a copyright more restrictive than CC-BY, please include a copy, in both the original language and English, as Supporting Information. If the original language is written in, non-Latin characters, for example Amharic, Chinese, or Korean, please use a file format that ensures these characters are visible

Response: ok we will provide All Questioners as additional information upon summation 

3. We note that you have indicated that data from this study are available upon request. PLOS only allows data to be available upon request if there are legal or ethical restrictions on sharing data publicly

Response: no restriction on data we can attach as supplementary files.

4. Please state whether you validated the questionnaire before testing on study participants. Please provide details regarding the validation group within the methods section.

Response: Thank you very much we had validated all tool using Cronbach alpha and we had discussed in detail in the Methods section of the manuscript 

5. Thank you for stating the following in the Funding Section of your manuscript:

[Arba Minch University as a requirement for postgraduate studies supports this research

financially.] We note that you have provided funding information that is not currently declared in your Funding Statement. However, funding information should not appear in the Acknowledgments section or other areas of your manuscript. We will only publish funding information present in the Funding Statement section of the online submission form.

Response: Thank you very much we had removed it 

Point by point response to Reviewers

Question 1: background is low what is the importance, necessary, gap, and research question.

The structure paragraph is incorrect. Article structure is weak.

Response 1: We would like to say thank you very much for your invaluable comments and suggestions. We considered and modified and rewrote again background section based on your constructive issues, coherence, and comprehensibility of the manuscript 

Question 2: What delivery type with normal vaginal delivery or cesarean delivery? 

Response 2: those mothers who have a normal vaginal delivery

Question 3. exclusion criteria are not completed

Response 3: We amend it and corrected it accordingly. Those women who are seriously ill and unable to communicate during the data collection period were excluded 

Question 4. please explain more about validity and reliability of questionnaire

Response 4: We amend it and corrected it accordingly. After pre-testing the questionnaire, Cronbatch’s Alpha was calculated by using SPSS window version 25.0 to test internal consistency (reliability) of the item and Cronbatch’s Alpha greater than 0.7 was considered as reliable. Data were collected by trained midwives and nurses. During data collection regular supervision was done by the supervisors

Question 5. please edit your table? there are a ambitious table with lot of rows

Response 5: It was corrected according to your suggestion 

Question 6. some lines in table have to delete to clarify it.

Response 6. It was corrected accordingly 

Question 7. discussion is weak and underdeveloped and do not cover all finding such as associated factors.

Response 7. In discussion section, correction was made accordingly based on both reviewers comment and suggestion 

Question 8. the paragraph structure is incorrect.

Response 8. correction was made according to you nice comment and suggestion 

Question 9. Edite your words example South Africa (??)

Response 9. Words correction was made accordingly based on both reviewers comment and suggestion 

Question 10. The references are incomplete

 Response 10: Correction was made thanks in-depth, for your nice comment

---

## [Decision Letter · Decision Letter 1]

4 Sep 2020

PONE-D-20-18273R1

Utilization of Companionship during delivery and associated factors among women who gave birth at Arbaminch town public health facilities, southern Ethiopia

PLOS ONE

Dear Dr. Alemu,

Thank you for submitting your manuscript to PLOS ONE. After careful consideration, we feel that it has merit but does not fully meet PLOS ONE’s publication criteria as it currently stands. Therefore, we invite you to submit a revised version of the manuscript that addresses the points raised during the review process.

SPECIFIC ACADEMIC EDITOR COMMENTS: There is still a major concern that the manuscript was not properly copyedited before resubmission. The authors must hire someone to proof their manuscript before resubmission.

We look forward to receiving your revised manuscript.

Kind regards,

Frank T. Spradley

Academic Editor

PLOS ONE

Reviewers' comments:

Reviewer's Responses to Questions

**Comments to the Author**

1. If the authors have adequately addressed your comments raised in a previous round of review and you feel that this manuscript is now acceptable for publication, you may indicate that here to bypass the “Comments to the Author” section, enter your conflict of interest statement in the “Confidential to Editor” section, and submit your "Accept" recommendation.

Reviewer #1: (No Response)

2. Is the manuscript technically sound, and do the data support the conclusions?

Reviewer #1: Partly

3. Has the statistical analysis been performed appropriately and rigorously? 

Reviewer #1: I Don't Know

4. Have the authors made all data underlying the findings in their manuscript fully available?

Reviewer #1: Yes

5. Is the manuscript presented in an intelligible fashion and written in standard English?

Reviewer #1: No

6. Review Comments to the Author

Reviewer #1: I would recommend a round of intensive copy-editing to improve the readibility of the paper. This was a comment made in the previous round of feedback and this challenge persists.

7. PLOS authors have the option to publish the peer review history of their article (what does this mean?). If published, this will include your full peer review and any attached files.

Reviewer #1: No

---

## [Author Response · Author response to Decision Letter 1]

16 Sep 2020

Author’s Point-by-Point Response to the Reviewer's and Editors Reports

Title: Utilization of Companionship during delivery and associated factors among women who gave birth at Arba Minch town public health facilities, southern Ethiopia: A cross-sectional study

Corresponding author: Biresaw Wassihun /bireswas@gmail.com

Authors

1. Kassaw Beyene

2. Gebresilasea Gendisha Ukke

3. Biresaw Wassihun

Manscurpuit number: PONE-D-20-18273

Journal: Plos one

Article type: Research article

Point by point response to Reviewers and Editors

First of all, the authors would like to thank Plos one Journal editors and the respective reviewers for reviewing our manuscript and providing the necessary comments to be corrected. As per the comments given, we have made corrections point by point to comment. The authors tried to answer all the issues raised by editorial team and reviewers. 

Point by point response to Editor

1. Please ensure that your manuscript meets PLOS ONE's style requirements, including those for file naming."" 

Response: Thank you very much we had apply journal requirement 

Response: ok we will provide 

Point by point response to Reviewers

Question 1. Reviewer #1: I would recommend a round of intensive copy-editing to improve the readibility of the paper. This was a comment made in the previous round of feedback and this challenge persists

Response: thank you for your nice comment. All of the comments was edited 

Question 2. Please state whether you validated the questionnaire before testing on study participants. Please provide details regarding the validation group within the methods section.

Response: Thank you very much we had validated all tool using Cronbach alpha . Before actual data collection occurred two-day training was provided for data collectors and the supervisor about techniques of data collection and briefed on each question included in the data collection tool. The pretest was done on 5% (21) of mothers receiving care in a health center that was not included in the study before the actual study period. After pre-testing the questionnaire, Cronbach's Alpha was calculated by using SPSS window version 25.0 to test internal consistency (reliability) of the item, and Cronbach's Alpha greater than 0.7 was considered as reliable. Data were collected by trained midwives and nurses. During data collection, regular supervision was done by the supervisors

Question 3: background is low what is the importance, necessary, gap, and research question.

The structure paragraph is incorrect. Article structure is weak.

Response : We would like to say thank you very much for your invaluable comments and suggestions. We considered and modified and rewrote again background section based on your constructive issues, coherence, and comprehensibility of the manuscript 

Question 5: What delivery type with normal vaginal delivery or cesarean delivery? 

Response : those mothers who have a normal vaginal delivery

Question 6. exclusion criteria are not completed

Response : We amend it and corrected it accordingly. Those women who are seriously ill and unable to communicate during the data collection period were excluded 

Question 7. please edit your table? there are a ambitious table with lot of rows

Response : It was corrected according to your suggestion 

Question 8. some lines in table have to delete to clarify it.

Response . It was corrected accordingly 

Question 9. discussion is weak and underdeveloped and do not cover all finding such as associated factors.

Response . In discussion section, correction was made accordingly based on both reviewers comment and suggestion 

Question 10. Edite your words example South Africa (??)

Response 9. Words correction was made accordingly based on both reviewers comment and suggestion

---

## [Editor Report · Decision Letter 2]

23 Sep 2020

Utilization of Companionship during delivery and associated factors among women who gave birth at Arbaminch town public health facilities, southern Ethiopia

PONE-D-20-18273R2

Dear Dr. Alemu,

We’re pleased to inform you that your manuscript has been judged scientifically suitable for publication and will be formally accepted for publication once it meets all outstanding technical requirements.

Kind regards,

Frank T. Spradley

Academic Editor

PLOS ONE

---

## [Editor Report · Acceptance letter]

25 Sep 2020

PONE-D-20-18273R2 

Utilization of Companionship during delivery and associated factors among women who gave birth at Arba Minch town public health facilities, southern Ethiopia   

Dear Dr. Alemu:

I'm pleased to inform you that your manuscript has been deemed suitable for publication in PLOS ONE. Congratulations! Your manuscript is now with our production department. 

Kind regards, 

on behalf of

Dr. Frank T. Spradley 

Academic Editor

PLOS ONE